# Whole Genome Sequence Analysis of Multidrug Resistant *Escherichia coli* and *Klebsiella pneumoniae* Strains in Kuwait

**DOI:** 10.3390/microorganisms10030507

**Published:** 2022-02-25

**Authors:** Ola H. Moghnia, Nourah A. Al-Sweih

**Affiliations:** Department of Microbiology, Faculty of Medicine, Kuwait University, Al-Safat 24923, Kuwait; ola.maghnia@ku.edu.kw

**Keywords:** whole genome sequence, resistome, *Klebsiella pneumoniae*, *Escherichia coli*

## Abstract

The spread of carbapenem-resistant *Escherichia coli* and *Klebsiella pneumoniae* is a global concern. The management of infections caused by multidrug resistance (MDR) isolates poses substantial clinical challenges in both hospitals and communities. This study aimed to investigate the genetic characteristics and variations of MDR *E. coli* and *K. pneumoniae* isolates. Bacterial identification and antibiotic susceptibility testing against 19 antibiotics were performed by standard methods. Whole genome sequencing (WGS) was carried out on eight carbapenem-resistant isolates using an Illumina MiSeq platform. The assembled draft genomes were annotated, then sequences were blasted against antimicrobial resistance (AMR) genes database. WGS detected several resistance genes mediating the production of β-lactamases, including carbapenems and extended-spectrum β-lactamase genes as *(bla_OXA-1/-48_*, *bla*_KPC-2/-29_, *bla*_CMY-4/-6_, *bla*_SHV-11/-12_, *bla*_TEM-1_, *bla*_CTX-M-15_, *bla*_OKP-B_, *bla*_ACT_ and *bla*_EC_). Furthermore quinolone resistance including *oqxA/oqxB*, *aac(6′)-Ib-cr5*, *gyrA_D87N*, *gyrA_S83F*, *gyrA_S83L*, *parC_S80I*, *parE_S458A*, *parE_I355T*, *parC_S80I*, and *qnrB1*. In addition to aminoglycoside modifying enzymes genes (*aph(6)-Id*, *aph(3*″*)-Ib*, *aac(3)-IIa*, *aac(6′)-Ib*, *aadA1*, *aadA2* and *aadA5*), trimethoprim-sulfamethoxazole (*dfrA12/A14/A17 and sul1/sul2*), tetracycline (*tetA* and *tetB*), fosfomycin (*fosA and uhpT_E350Q*) resistance genes, while other genes were detected conferring chloramphenicol (*floR*, *catA2*, and efflux pump *cmIA5*), macrolides resistance (*mph(A) and erm(B)*, and quaternary ammonium efflux pump *qacEdelta.* Bleomycin and colistin resistance genes were detected as *ble* and *pmrB_R256G*, respectively. Comprehensive analysis of MDR strains provided by WGS detected variable antimicrobial resistance genes and their precise resistance mechanism. WGS is essential for control and prevention strategies to combat the growing threat of AMR and the implementation of multifaceted interventions are needed.

## 1. Introduction

The escalating burden of antimicrobial resistance worldwide is substantial and is likely to grow [1]. The situation is complicated, since the development of resistance among Gram-negative bacteria is more rapid than Gram-positive bacteria [2], particularly antimicrobial-resistance strains (namely extended spectrum beta-lactamase [ESBL] and carbapenemase producers). The emergence of multidrug resistance (MDR) Enterobacterales such as *Klebsiella pneumoniae* and *Escherichia coli* is a well-known global health concern and possesses numerous resistance genes in its genome [3]. In healthcare settings, carbapenems are the drug of choice and are the most potent group of β-lactam antibiotics used to treat serious infections caused by ESBL-producing Enterobacterales and AmpC β-lactamase producers. They are frequently used for empiric therapy of life-threatening infections, such as sepsis [4,5]. The rising carbapenem resistance phenomenon in *K. pneumoniae* and *E. coli* is of particular concern, as this may lead to almost untreatable community-acquired infections [6]. The emergence and spread of MDR *K. pneumoniae* and *E. coli* isolates complicate the treatment of severe infections and threaten to create strains of bacteria resistant to currently available antimicrobial agents such as the β-lactam antibiotics [7]. The intensive use of antimicrobials in current practice may explain the expedition in developing and disseminating genes that encode resistance to β-lactam and other classes of antibiotics in *K. pneumoniae* and *E. coli*. Methods of discriminating and characterizing different *K. pneumoniae* and *E. coli* isolates are essential to targeting infection control resources. Several studies have been conducted in other countries detecting the prevalence of *E. coli* and *K. pneumoniae* carbapenem resistance. Prevalence rates ranged between 1 and 4% in Lebanon [8], 3% in Syria [9], 4% in Iraq [10], 22.5% in Jordan [11] and 2% or less in African Arab countries such as Algeria, Libya, Morocco, Mauritania and Tunisia. However, reports from Egypt revealed the highest prevalence rate of 28% among *E. coli* and *K. pneumoniae* isolates [12,13]. A study showed increased prevalence rates of carbapenem resistance among *K. pneumoniae* than in *E. coli* isolates in Saudi Arabia [14]. In Kuwait, carbapenem resistance was detected in 3.4% among *K. pneumoniae* and *E. coli* strains isolated from the community [15]. To improve our understanding of transmission dynamics and antimicrobial resistance (AMR) gene diversity in outbreaks, it is recommended to use whole genome sequencing (WGS) of bacterial isolates, which is now well placed to become a gold standard in bacterial typing and to determine relatedness between strains, as it is highly discriminant in characterizing the complete genomic structure of isolates [16,17]. Advances in WGS assays have shown them to be a powerful tool for genotyping and the bacterial identification of clones in outbreak transmission with speed and depth of information [18]. Molecular dynamics reports of MDR isolates driving from the community at the whole genome level is limited. Previously, we identified the prevalence of genes mediating carbapenemase production in Enterobacterales isolates circulating in the community. We found that a relatively high number of carbapenem-resistant isolates harbored predominantly blaKPC among healthy individuals in community settings, which was uncommon in Kuwait and neighboring countries. In this study we sought to assess the use of whole genome sequencing (WGS) on selected *E. coli* and *K. pneumoniae* carbapenem-resistant isolates recovered from healthy individuals and patients, in order to understand their genomic diversity and genotypic presence of antimicrobial resistance.

## 2. Materials and Methods

### 2.1. Bacterial Identification and Antimicrobial Susceptibility

The study was carried out between May 2020 and October 2021 on eight MDR and confirmed carbapenem-resistant *E. coli* and *K. pneumoniae* isolates from rectal swabs recovered from healthy individuals and patients admitted to different hospitals in Kuwait. Representatives of carbapenem-resistant *E. coli* and *K. pneumoniae* isolates with carbapenemase-positive (*n* = 2) and carbapenemase-negative (*n* = 2) for each were chosen to gain further insight into the origin of genetic diversity and investigate resistance mechanisms. Sample collection and isolation methods were previously described [19,20]. Isolates were identified by VITEK 2 ID system (bioMérieux, Marcy-l’Etoile, France) and antimicrobial susceptibility testing was carried out to determine the minimum inhibitory concentration (MIC) in μg/mL against 19 antibiotics using the Epsilometer test (*E* test) (bioMérieux, Marcy-l’Etoile, France), except for colistin, which was tested by agar dilution method as described previously [20]. Non-susceptible isolates to either one or more of the tested carbapenems with MIC of >0.5 µg/mL for ertapenem or >1 µg/mL for both imipenem and meropenem were selected for analysis. Antibiotic susceptibility results were interpreted according to the Clinical and Laboratory Standards Institute, 2020 breakpoints [21].

### 2.2. Whole Genome Sequencing

Genomic DNA from *E. coli* and *K. pneumoniae* were extracted using QIAamp^®^ DNA Mini Kit (Qiagen, Hilden, Germany) according to the manufacturer’s instructions. For each isolate, genomic DNA was quantified by the NanoDrop-800 spectrophotometer (Thermo Fisher Scientific, Wilmington, NC, USA) following the manufacturer’s protocol. DNA extracts were fully sequenced by whole genome paired-end sequencing, Illumina data (2 × 150 bp reads), including whole genome reconstruction, annotation, and antimicrobial resistance gene typing using the MiSeq sequencer (Illumina, San Diego, CA, USA). Accordingly, DNA libraries were prepared using Nextera XT library (Illumina, San Diego, CA, USA) targeted for each genome with 1 ng genomic DNA per the manufacturer’s recommendations. Briefly, tagment genomic DNA was simultaneously fragmented and then tagged with adapter sequences in a single step using Nextera transposome (Nextera XT DNA Library Preparation Kit, Illumina, San Diego, CA, USA). Tagmented DNA was then amplified using a limited-cycle (12-cycle) PCR program. To purify the library DNA, amplified DNA was cleaned up with AMPure XP beads. Then, Nextera libraries were quantified using Qubit and the size profile was determined on Agilent Technology 2100 Bioanalyzer using a high-sensitivity DNA chip (Agilent Technologies, Waldbronn, Germany). Various libraries can be sequenced with average fragment sizes that were generated from 828 to 1433 bases. Selected libraries for sequencing were normalized to 1 nM and pooled. Pooling libraries combined equal volumes of normalized libraries in a single tube. Then, 1 nM pooled library was diluted and heat-denatured before loading libraries for the sequencing run on a MiSeq sequencer (MiSeq reagent kit V2-300 cycles).

### 2.3. Bioinformatics Analysis

Paired-end reads from each *E. coli* and *K. pneumoniae* isolates were adapted, trimmed and quality-controlled using *BBduk* tool of the *BBTools* package (v38.59) (available online at https://jgi.doe.gov/data-and-tools/software-tools/bbtools/bb-tools-user-guide/, accessed on 2 September 2021) to remove adapters, low-quality bases and ambiguous reads. Trimmed raw data were further assembled de novo using Spades (v3.13.1) algorithm to create a draft genome sequence for each isolate [22] and the data quality was checked with FastQC tool (v0.11.7) (available online at https://www.bioinformatics.babraham.ac.uk/projects/fastqc/, accessed on 2 September 2021). The assembled draft genomes were annotated by Prokka (v1.13.3) and CDS and protein sequences were extracted [23]. The WGS data were uploaded into the center for Genomic Epidemiology (available online at https://www.genomicepidemiology.org/services/index.html, accessed on 11 February 2022 and at https://pathogen.watch/, accessed on 11 February 2022) to detect virulence genes. The presence of insertion sequences was detected using ISFinder (available online at https://www-is.biotoul.fr/blast.php/, accessed on 11 February 2022) to confirm their identity. The sequence types of isolates were determined based on WGS data (available online at https://cge.cbs.dtu.dk/services/mlst/, accessed on 11 February 2022).

### 2.4. Antimicrobial Resistance Gene Typing

Concurrently, each draft genome was screened for the presence of AMR genes. The most updated database of AMR genes downloaded from NCBI National Database of Antibiotic-Resistant Organisms (available online at https://www.ncbi.nlm.nih.gov/pathogens/antimicrobial-resistance/, accessed 2 September 2021) was used as the reference for identifying the resistance genes present in the isolates. The scaffolds were blasted against all classes of AMR reference genes to infer the presence of all types of antibiotics. Sequenced raw data were blasted against retrieved genes using Standalone Blast (blastn, available online at https://blast.ncbi.nlm.nih.gov/Blast.cgi, accessed on 2 September 2021) [24].

## 3. Results

### 3.1. Characteristics of Participants 

Four *E. coli* isolates were obtained from three females and one male. Of these, three Kuwaiti patients were admitted to three different hospitals, including Ibn Sina hospital (IBS) and Babtain hospital (Bab), located in the Capital (Al-Asimah) Governorate, in addition to Mubarak hospital (MK) in Hawali Governorate. The fourth isolate belonged to the healthy Filipino participant who lived in Farwaniya Governorate. However, four *K. pneumoniae* isolates were obtained from three males and one female. Two isolates belonged to Canadian and Kuwaiti patients who were admitted to MK in Hawali Governorate and Farwaniya hospital (FA) in Farwaniya Governorate, respectively. In addition, two isolates from healthy Indian participants who lived in Hawali and Farwaniya Governorates. All participants were aged between 1–80 years old (Table 1).

### 3.2. Genome Accession Numbers

Raw genome data were submitted to the National Center for Biotechnology Information (NCBI) (available online at https://www.ncbi.nlm.nih.gov/sra/, accessed on 11 July 2021). The assembled genomic sequences of *K. pneumoniae* and *E. coli* isolates were deposited under the BioProjects (accession number PRJNA632581 and accession number PRJNA630112), respectively.

### 3.3. Antibiotic Sensitivity Test

Table 2 shows the antimicrobial susceptibility findings of eight isolates against 19 antimicrobial agents. All isolates were carbapenemase producers conferring resistant to ertapenem (*n* = 7), imipenem (*n* = 2), and meropenem (*n* = 2). A multi-drug resistance (MDR) profile, based on resistance to at least one agent in three or more antimicrobial classes was detected in all isolates. *E. coli* strains E1-112, E2-466, E3-471, and E4-485 were resistant to 8, 15, 13 and 17 antibiotics, respectively. While *K. pneumoniae* strains K1-245, K2-351, K3-441, and K4-500 were resistant to 3, 4, 13 and 12 antibiotics, respectively. The number of resistant isolates against 19 antibiotics are as following; β-lactams [ampicillin (*n* = 7), amoxicillin–clavulanic acid (*n* = 5), piperacillin (*n* = 5)]; monobactam [aztreonam (*n* = 5)]; cephalosporins [cephalothin (*n* = 6), cefepime (*n* = 4), cefotaxime (*n* = 6), cefoxitin (*n* = 4), ceftazidime (*n* = 6), ceftriaxone (*n* = 5) and cefuroxime (*n* = 6)]; ciprofloxacin (*n* = 3)]; colistin (*n* = 1); aminoglycosides [amikacin (*n* = 2) and gentamicin (*n* = 2)] and tetracycline (*n* = 6).

### 3.4. Whole Genome Sequencing and In Silico Data Analysis

The isolates’ final assembly, obtained by WGS, ranged from 52 to 138 contigs of >500 bps/sample in *E. coli* isolates with N50 values of between 124,277 and 560,222. A total of 52, 138, 115 and 107 contigs representing 4,745,810; 5,236,280; 5,369,672 and 5,463,758 bases (56% and 50% G + C ratio; N50 = 560,222, 124,277, 183,231 and 186,146) were obtained from assembled sequences of *E. coli* strains E1-112, E2-466, E3-471, and E4-485, respectively. In addition, 4417, 4845, 5015, 5149 CDS; 2, 3, 2, 3 rRNA; 69, 67, 73, 71 tRNA and 4489, 4916, 5091, 5224 mRNA, respectively, were annotated for final contigs.

The isolates’ final assembly ranged from 69 to 106 contigs of >500 bps/sample in *K. pneumoniae* isolates with N50 values of 139,608 and 375,813. A total of 106, 80, 71 and 69 contigs representing 5,536,425; 5,4243,28, 5,337,760 and 5,498,821 bases (56% and 57% G + C ratio; N50 = 139,608, 272,188, 375,813 and 294,715) were obtained from assembled sequences of *K. pneumoniae* strains K1-245, K2-351, K3-441, and K4-500, respectively. Moreover, 5218, 5015, 4949, 5156 CDS; 2, 2, 1, 1 rRNA; 68, 74, 69, 70 tRNA; and 5289, 5092, 5020, 5228 mRNA were annotated for final contigs. The main features of the *E. coli* and *K. pneumoniae* genome are shown in Table 3 and Table 4.

### 3.5. Antimicrobial Resistance Pattern

Resistome analysis revealed the presence of various resistance genes that are demonstrated in Table 5. *E. coli* isolates carried the following β-lactamase genes (E1-112; *bla_ACT_* for AmpC β-lactamse enzymes) (E2-466; *bla_EC_*, *bla_CTX-M-15_*, *bla_OXA_*_-*1*_); (E3-471; *bla_KPC-2_*, *bla_CMY-4_*, *bla_TEM_*) and (E4-485; *bla_EC_*, *bla_CMY-6_*, *bla_CTX-M-15_*). *K. pneumoniae* isolates harboured the following β-lactamase genes (K1-245; *bla _KPC-29_*, *bla_OXA-48_*); (K2-351; *bla_SHV-11_*); (K3-441; *bla_OKP-B_*, *bla_SHV-12_*) and (K4-500; *bla_SHV-11_*, *bla_TEM-1_*, *bla_CTX-M-15_*, *bla_OXA-1_*). The presence of β-lactamase genes was related with sequences encoding for resistance to other classes of antibiotics in all the isolates: Three *E. coli* isolates conferring dihydrofolate reductase genes *dfrA*, including *dfrA12/14/17* alleles encoded trimethoprim resistance enzyme. Six isolates (3 *E. coli* and 3 *K. pneumoniae*) harbored *sul1* and *sul2* genes encoded sulfonamide resistance. The *mph (A)* and *erm(B)* genes were found in 3 *E. coli* and 2 *E. coli* isolates, respectively, encoding macrolide resistance. Both *tetA*, *tetB* genes encoding tetracycline resistance were detected in 2 *K. pneumoniae* and 2 *E. coli*, respectively. Aminoglycoside-modifying enzymes were detected in isolates with the following genes *aph(6)-Id* (1 *E. coli* and 2 *K. pneumoniae*), *aph(3″)-Ib* (1 *E. coli and 2 K. pneumoniae*), *aac(6′)-Ib3* (1 *E. coli*), *aac(6′)-Ib* (1 *K. pneumoniae*), *aac(3)-IIa* (*1 E. coli*), *aadA1*(*1 K. pneumoniae*), *aadA2* (1 *E. coli*), and *aadA5* (1 *E. coli).* Quinolone resistance was detected in 8 distinct genes, *gyrA_D87N* (*2 E. coli and 1 K. pneumoniae*), *gyrA_S83F* (1 *K. pneumoniae*), *gyrA_S83L* (2 *E. coli*), *parC_S80I (2 E. coli)*, *parE_S458A* (*1 E. coli*), *parE_I355T* (1 *E. coli*), *parC_S80I* (2 *E. coli*), *qnrB1* (*1 K. pneumoniae).* For quinolone resistance, *aac(6′)-Ib-cr5* gene was detected in 1 *E. coli* isolate. Furthermore, 1 *E. coli* and *4 K. pneumoniae* isolates harbored multidrug efflux RND (Resistance-Nodulation-Division) transporter periplasmic adaptor subunit *oqxA* and multidrug efflux RND transporter permease subunit *oqxB*. For colistin resistance, *pmrB_R256G* gene was detected in 1 *K. pneumoniae* isolate. Six isolates (2 *E. coli* and 4 *K. pneumoniae*) carried *fosA* gene and 2 *E. coli* isolates carried *uhpT_E350Q* that conferred resistance to fosfomycin. Three *E. coli* isolates carried the *cyaA_S352T* gene that conferred resistance to fosmidomycin. Resistance genes *catA2* in 1 *K. pneumoniae* isolate and *floR* gene in 1 *E. coli* isolate were found to confer resistance to phenicol. The presence of *ble* gene detected in 1 *E. coli* isolate is responsible for bleomycin resistance. Three *E. coli* isolates carried quaternary ammonium compound efflux pump SMR transporter *qacE delta1*, and a single *K. pneumoniae* isolate harbored two efflux pumps: chloramphenicol efflux MFS transporter *cmlA5* gene and *qacE delta1*. Virulence genes were elucidated in *E.coli* isolates, E2-466 harbored enteroaggregative immunoglobulin repeat protein (*air)*, outer membrane hemin receptor *(chuA)*, *colicin ia (cia)*, salmonella HilA homolog *(eilA)*, siderophore receptor *(fyuA)*, glutamate decarboxylase *(gad)*, heat-resistant agglutinin *(hra)*, high molecular weight protein 2 non-ribosomal peptide synthetase *(irp2)*, capsule polysaccharide export inner-membrane protein *(kpsE)*, polysialic acid transport protein; group 2 capsule *(kpsMII_K5)*, tellurium ion resistance protein *(terC)*, and outer membrane protein complement resistance (*traT*). E3-471 isolate harbored the following genes: *air*, *chuA*, *colicin ib (cib)*, *gad*, long polar fimbriae *(lpfA)*, *terC*, *traT*, *uropathogenic specific protein (usp)*, and *fimbrial protein (yfcV)*. E4-485 isolate carried the following: (*air*, *chuA*, *eilA*, *fyuA*, *gad*, *irp2*, increased serum survival *(iss)*, *aerobactin synthetase (iucC)*, ferric aerobactin receptor *(iutA)*, capsule polysaccharide export inner-membrane protein *(kpsE)*, *lpfA*, outer membrane protease *(protein protease 7) (ompT)*, major pilin subunit F16 *(papA_fsiA_F16)*, outer membrane usher P fimbriae *(papC)*, *secreted autotransporter toxin (sat)*, *plasmid-encoded enterotoxin (senB)*, and iron transport protein *(sitA)*, *terC*, and *traT*). All *K. pneumoniae* isolates harbored wzi104, capsular locus (K locus) variant KL51, and (O locus) O1v2. Sequence types for *E. coli* isolates *(*E1-112, E2-466, E3-471 and E4-485*)* belonged to ST499, ST405, ST354 and ST69, respectively. Additionally, *K. pneumoniae* isolates *(*K1-245, K2-351, K3-441 and K4-500) belonged to ST3495, ST25, ST4743 and ST37, respectively. The insertion sequences as ISEc19, IS609, IS4, ISEC46 were detected in *E*. *coli* isolates and ISKpn74, ISKpn78, ISEc23 and ISKpn1 were detected in *K. pneumoniae* isolates.

## 4. Discussion

*E. coli* and *K. pneumoniae* are important human pathogens that cause a wide spectrum of clinical diseases. To our knowledge, this is the first study to report the WGS analysis of asymptomatic carriers of MDR isolates in community settings in Kuwait and neighboring countries. Exploring the presence of AMR colonization is exceedingly essential, as these carriers may act as reservoirs for transmission of MDR bacteria. Moreover, the numbers are underestimated due to the absence of surveillance studies and efficient screening methods of individuals and focus only on patients admitted to healthcare settings. The advantages of WGS over the traditional sequencing technology is the ability to generate millions of reads in a single run with low costs. In the present study, the wide distribution of antibiotic resistance genes among isolates, including β-lactamase genes, ESBL and other genes conferring resistance to fosfomycin, aminoglycoside, sulfonamide, quinolone, phenicol, tetracycline, fosmidomycin, trimethoprim, macrolide, bleomycin, and colistin, in addition to several efflux pumps. Interestingly, *K. pneumoniae* and *E. coli* isolates showed elevated resistance rates to nearly all the β-lactam tested and other multiple antibiotics. All isolates were MDR phenotypes that carried at least one β-lactamase gene and the MICs of carbapenem antibiotics supported the presence of resistance genes of these antibiotics. The prevalence of Gram-negative pathogens has practical importance, especially for treatment options. In line with the findings of an Italian study, β-lactamase characterization of carbapenem-resistant *K. pneumoniae* isolates demonstrated the preponderance of carbapenemase production as *K. pneumoniae* carbapenemase (KPC) variants (*bla*KPC-2, *bla*KPC-3) and *bla*OXA. In addition, ESBLs, which include the Cefotaximase-Munich (CTX-M), Temoneria (TEM), and Sulfhydryl Variable (SHV) families were detected [25]. Our findings showed that the most prevalent ESBLs belong to *bla*SHV-11/12, *bla*TEM-1, *bla*CTX-M-15, *bla*CMY-6 genes. Those isolates also co-harbored carbapenemases as well *bla*KPC variants, *bla*KPC-2 and *bla*KPC-29 in addition to oxacillinase carbapenemase (OXA) variants, as *bla*OXA-48 and *bla*OXA-1 were detected. WGS confirmed the presence of KPC variants in Kuwait hospitals and community as one *E. coli* isolate with ST354 obtained from a Kuwaiti patient admitted to Mubarak hospital. The other isolate was *K. pneumoniae* with ST3495, obtained from a healthy Indian; they both live in Hawali Governorate, unlike previous publications that reported the predominance of OXA and New Delhi metallo-β-lactamas (NDM) variants in our region. It should be noted that *K. pneumoniae* disseminates easily via person-to-person contact, and its pathogenicity is enhanced by the easy acquisition of β-lactamase encoding genes. Therefore, the co-evolution of carbapenem-resistant strains is potentially the most worrying possibility due to the emergence of invasive *K. pneumoniae* infections, leading to treatment challenges.

Another interesting finding in our study was detecting a variety of resistance genes, highlighting the presence of *sul* gene in 7 isolates and *fosA* gene in 6 isolates. Our findings parallel a study that reported the increased prevalence rates of *sul* variants, primarily in Gram-negative bacteria, including *E. coli* strains isolated worldwide [26]. However, *fosA* gene is frequently found in Gram-negative species, including *K. pneumoniae* and *P. aeruginosa*, contributing to intrinsic fosfomycin resistance and it was rarely identified in *E. coli*. This may be due to the presence of *fosA* gene on the chromosome in many Gram-negative species, whereas it rarely exists on the *E. coli* chromosome [27]. However, our findings showed that *fosA* gene was detected in all tested *K. pneumoniae* and two *E. coli* isolates; one of these *E. coli* isolates is a KPC-producing carbapenemase. A study reported that *fosA*6 detected in a fosfomycin-resistant *E. coli* strain was likewise mobilized from the chromosome of *K. pneumoniae,* depending on its high degree of resemblance with *fosA*^KP^ and its position on a transposon [28]. Therefore, chromosomal *fosA* genes in Gram-negative bacterial species may act as a reservoir of fosfomycin resistance in species without *fosA* gene, such as *E. coli* [29]. 

Genomic data can be applied to trace and identify unanticipated modes of transmission and types of resistance genes. In our study, colistin resistance was conferred in an MDR *K. pneumoniae* isolate harbored *pmrB_R256G* gene obtained from a Kuwaiti patient. Resistance to colistin was not detected phenotypically and the isolate was within the susceptible MIC range for colistin; this may enhance the dissemination of such resistant genes silently in the hospital, as well as in the community. Moreover, it is known that *K. pneumoniae* has a high degree of environmental stability and survival in hospital environments and utensils, despite rigorous cleaning and decontamination procedures [30]. If we had such information, this asymptomatic patient could have been subjected to further strict surveillance cultures to detect potential sources of colonization and/or might be placed on contact isolation to cease any probable silent transmission chain that may occur in the hospital. Aminoglycoside resistance rates in *E. coli* and *K. pneumoniae* have been described in many countries, leading to global concerns [31,32]. The mechanism of resistance to aminoglycosides mainly occurs due to the presence of aminoglycoside-modifying enzymes (AMEs) and their different ability to modify aminoglycosides. These enzymes include acetyltransferases (AACs), nucleotidyltransferases (ANTs), and phosphotransferases (APHs) were described in Enterobacterales [33]. In our findings, MICs for gentamicin and/or amikacin showed high-level resistance in 3 isolates expressing various aminoglycosides gene as E4-485, *aac(6′)-Ib3* gene; E2-466, *aad* and *aac(3)-IIa* genes and K3-441, *aac(6′)-Ib* and *aadA1* genes. However, 3 isolates were susceptible to aminoglycosides tested found to carry variety of aminoglycosides resistance genes including E3-471, *aph(6)-Id*, *aph(3″)-Ib*, *aadA2*; K1-245, *aph(6)-Id*, *aph(3″)-Ib* and K4-500, *aph(6)-Id*, *aph(3″)-Ib.* It should be noted that other types of aminoglycosides such as streptomycin, kanamycin, netilmicin, and tobramycin were not tested in our study, which may cause the presence of resistance genes in those isolates. Quinolone resistance was detected in 4 isolates; of these 3 isolates showed elevated MIC value to ciprofloxacin (MIC > 23 µg/mL) and a great variety of quinolone-resistance determining regions (QRDRs) genes were detected. DNA gyrase (gyrA) mutations encoding the amino acid substitutions S83L, 83F and D87N were detected in 2 *E. coli* and 1 *K. pneumoniae* isolates. These substitutions have been associated with quinolone resistance for *E. coli* and confer higher resistance levels than any other substitutions in the QRDRs, similar to that reported elsewhere [34], in addition to mutations causing the substitution in topoisomerase IV, *parC_S80I*, *parE_S458A* and *parE_I355T.* Another interesting finding is the presence of ciprofloxacin-sensitive *K. pneumoniae* that harboured plasmid-mediated quinolone resistance (PMQR) gene and expressed by the mediation of *qnr* gene, namely as *qnrB1*. This gene produces a pentapeptide protein that protects targeted enzymes from the effect of quinolones. Additionally, the presence of another method involves a mutation of *aac(6′)-Ib-cr5* gene in one *E. coli* isolate, which confers resistance to quinolones by acetylation. A similar finding to that described by Magesh et al., 2011 reported the presence of multidrug-resistant *K. pneumoniae* isolates harboring *aac(6′)-1b-cr* mutant gene in India [35]. The detection of efflux pumps belonging to *oqxA* and *oqxB* in our isolates may explain the mechanism of resistance. It has been reported as one of the main mechanisms of plasmid-mediated quinolone resistance that reduces the concentrations of quinolones intracellularly [36]. All these detected mechanisms are considered major contributors to the intrinsic and acquired multidrug resistance in Gram-negative bacteria. Our data showed that WGS determine accurately the exact mechanisms of antibiotic resistance in Enterobacterales. WGS platforms are rapidly spreading in clinical diagnostic laboratories that make genomic data more accessible and easily used in routine clinical settings, due to the lowering of turnaround time. Updates in DNA sequence techniques, plasmid conjugation and gene cloning should greatly improve our understanding of the spread and evolution of resistant strains. A limitation of the study was the requirement of expertise for bioinformatics analysis of data.

## 5. Conclusions

Our findings showed that whole genome sequencing and its in-depth analysis can accurately reconstruct the molecular characterization of isolates. The global spread of MDR *K. pneumoniae* and *E. coli* is a worrying phenomenon and close inspection to avoid their spread is strongly warranted. The clinical application of WGS could improve the surveillance of alert MDR pathogens by overcoming the limitation of analyzing only a small part of the genome and providing more rapid management and controlling the emergence of new antibiotic-resistant strains and their evolution. In view of the changing epidemiology of MDR clones, as demonstrated in this study and other reports in the literature, it is recommended in the future to perform active surveillance and to monitor any changes in resistant patterns, carriage of genes for virulence factors and prevalent clones.

## Figures and Tables

**Table 1 microorganisms-10-00507-t001:** Demographic data of *Escherichia coli* and *Klebsiella pneumoniae* isolates obtained from patients and healthy participants.

Strain ID	Isolates	Population	Nationality	Age Group	Gender	Governorate	Accession Number
E1-112	*E. coli*	Healthy	Filipino	40–49	F	Farwaniya	SRX8356292
E2-466	*E. coli*	Patient	Kuwaiti	50–59	M	Capital/(IBS)	SRX8356293
E3-471	*E. coli*	Patient	Kuwaiti	70–79	F	Hawali/(MK)	SRX8356294
E4-485	*E. coli*	Patient	Kuwaiti	20–29	F	Capital/(Bab)	SRX8356295
K1-245	*K. pneumoniae*	Healthy	Indian	29–39	M	Hawali	SRX8344629
K2-351	*K. pneumoniae*	Healthy	Indian	29–39	F	Farwaniya	SRX8344630
K3-441	*K. pneumoniae*	Patient	Canadian	80–89	M	Hawali/(MK)	SRX8344631
K4-500	*K. pneumoniae*	Patient	Kuwaiti	1–9	M	Farwaniya/(FA)	SRX8344632

IBS: Ibn Sina hospital; MK: Mubarak hospital; Bab: Babtain hospital; FA: Farwaniya hospital.

**Table 2 microorganisms-10-00507-t002:** Resistance pattern of *Escherichia coli* and *Klebsiella pneumoniae* isolates against selected antimicrobial agents.

Antimicrobial Agents	Bacterial Isolates (Minimum Inhibitory Concentrations Breakpoints in µg/mL)							
	E1-112	E2-466	E3-471	E4-485	K1-245	K2-351	K3-441	K4-500
Amikacin	3	6	2	**>256**	1	1.8	**32**	4
Amoxacillin-Clavulinic acid	**32**	**16**	**32**	**32**	3	1.5	8	**>256**
Ampicillin	**>256**	**>256**	**>256**	**>256**	12	**48**	**>256**	**>256**
Aztreonam	0.125	**>256**	**48**	**64**	0.032	0.047	**>256**	**>256**
Cefepime	0.094	**64**	2	**>256**	0.047	0.047	**3**	**24**
Cefotaxime	0.094	**32**	**32**	**>256**	0.064	**4**	**24**	**32**
Cefoxitin	**>256**	**24**	**>256**	**>256**	3	4	4	8
Ceftazidime	1	**64**	**24**	**>256**	0.19	**16**	**>256**	**128**
Ceftriaxone	0.75	**>256**	**64**	**>256**	0.047	0.064	**32**	**>256**
Cefuroxime	**16**	**>256**	**>256**	**>256**	2	2	**48**	**>256**
Cephalothin	**>256**	**>256**	**>256**	**>256**	4	4	**>256**	**>256**
Ciprofloxacin	0.023	**32**	**32**	0.006	0.032	0.064	**32**	0.75
Colistin	0.25	0.25	0.25	0.125	0.75	0.75	0.19	0.38
Ertapenem	**1**	**3**	**12**	**4**	**1**	0.047	**1.5**	**4**
Gentamicin	1.5	**16**	1.5	**1024**	0.19	0.38	0.5	1.5
Imipenem	**6**	0.25	1	**3**	0.25	0.125	0.25	0.25
Meropenem	0.25	0.125	0.75	**2**	0.19	3	0.047	0.19
Piperacillin	1.5	**>256**	**>256**	**>256**	3	4	**>256**	**>256**
Tetracycline	**12**	**>256**	**>256**	**>256**	4	3	**48**	**128**

**Bold** indicates resistance.

**Table 3 microorganisms-10-00507-t003:** The summary statistics of the assembled draft genomes of *Escherichia coli and Klebsiella pneumoniae* isolates.

Genomic Data	*E. coli* Isolates								*K. pneumoniae* Isolates							
	E1-112		E2-466		E3-471		E4-485		K1-245		K2-351		K3-441		K4-500	
Raw reeds																
Total Sequences	5,707,885		4,791,414		4,590,486		5,474,281		5,266,901		6,780,089		5,566,937		4,870,982	
Sequence length	150		150		150		150		150		150		150		150	
QCed reads																
Total Sequences	5,588,157		4,679,003		4,490,667		5,345,636		5,157,922		6,632,494		5,452,319		4,758,468	
percent recovery	97.9		97.7		97.8		97.7		97.9		97.8		97.9		97.7	
Sequence length	50–140		50–140		50–140		50–140		50–140		50–140		50–140		50–140	
% GC	56		50		50		50		56		57		57		57	
Assembled draft genomes																
	bp	*n*	bp	*n*	bp	*n*	bp	*n*	bp	*n*	bp	*n*	bp	*n*	bp	*n*
Average contig size	91,266		37,944		46,693		51,063		52,230		67,804		75,180		79,693	
Largest contig	2,195,272		379,433		607,340		419,025		321,557		1,510,665		979,843		512,237	
Sum of base pairs	4,745,810	52	5,236,280	138	5,369,672	115	5,463,758	107	5,536,425	106	5,424,328	80	5,337,760	71	5,498,821	69
N50	560,222	2	124,277	13	183,231	9	186,146	10	139,608	14	272,188	5	375,813	5	294,715	7
N60	450,229	3	102,180	18	136,220	13	155,202	14	116,907	18	233,142	7	288,935	7	257,159	9
N70	344,880	4	79,987	23	124,445	17	95,100	19	87,397	24	195,143	9	204,896	9	174,886	12
N80	259,295	5	52,823	32	85,546	22	86,334	25	57,439	31	141,140	12	171,091	12	124,552	16
N90	136,066	8	29,561	45	38,747	32	48,438	33	31,232	44	101,299	17	83,691	16	86,473	21
N100	502	52	500	138	524	115	503	107	510	106	501	80	503	71	513	69
N_count	390		575		287		1358		1158		881		196		1458	
Gaps	4		6		3		14		12		9		2		15	

**Table 4 microorganisms-10-00507-t004:** Annotation of draft genome of *Escherichia coli and Klebsiella pneumoniae* isolates.

Genomic Data	*E. coli* Isolates				*K. pneumoniae* Isolates			
	E1-112	E2-466	E3-471	E4-485	K1-245	K2-351	K3-441	K4-500
contigs/scaffolds	52	138	115	107	106	80	71	69
bases	4,745,810	5,236,280	5,369,672	5,463,758	5,536,425	5,424,328	5,337,760	5,498,821
gene	4489	4916	5091	5224	5289	5092	5020	5228
CDS *	4417	4845	5015	5149	5218	5015	4949	5156
tRNA *	69	67	73	71	68	74	69	70
mRNA *	4489	4916	5091	5224	5289	5092	5020	5228
rRNA *	2	3	2	3	2	2	1	1
tmRNA *	1	2	1	1	1	1	1	1
CRISPR *	–	1	1	2	–	–	–	–

***** CDS coding sequence, tRNA: transfer RNA; mRNA: messenger RNA; rRNA: ribosomal RNA; tmRNA: transfer-messenger RNA; CRISPR: Clustered Regularly Interspaced Short Palindromic Repeats.

**Table 5 microorganisms-10-00507-t005:** Types of resistance genes and mechanisms of *Escherichia coli* and *Klebsiella pneumoniae* isolates to different antimicrobial agents detected by WGS.

Resistance to Antibiotics	*Escherichia coli* Isolates				*Klebsiella pneumoniae* Isolates			
	Antimicrobial Resistance Genes							
	E1-112	E2-466	E3-471	E4-485	K1-245	K2-351	K3-441	K4-500
Beta-lactam	*blaACT*	*blaEC blaCTX-M-15 blaOXA-1*	*blaKPC-2 blaCMY-4 blaTEM*	*blaEC blaCMY-6 blaCTX-M-15*	*blaKPC-29 blaOXA-48*	*blaSHV-11*	*blaOKP-B blaSHV-12*	*blaSHV-11 blaTEM-1 blaCTX-M-15 blaOXA-1*
Fosfomycin	*fosA uhpT_E350Q*		*fosA* *uhpT_E350Q*		*fosA*	*fosA*	*fosA*	*fosA*
Aminoglycoside		*aadA5* *aac(3)-IIa*	*aph(6)-Id aph(3″)-Ib aadA2*	*aac(6′)-Ib3*	*aph(6)-Id aph(3″)-Ib*		*aac(6′)-Ib aadA1*	*aph(6)-Id*, *aph(3″)-Ib*
Sulfonamide		*sul1*	*sul2*, *sul1*	*sul1*	*sul1*		*sul1*	*sul2*
Quinolone	*oqxA* *oqxB*	*aac(6′)-Ib-cr5*			*oqxA* *oqxB*	*oqxB17 oqxA10*	*oqxA* *oqxB*	*oqxA* *oqxB*
		*gyrA_D87N gyrA_S83L parC_S80I parE_S458A*	*gyrA_D87N gyrA_S83L parE_I355T parC_S80I*				*gyrA_D87N gyrA_S83F*	*qnrB1*
Phenicol			*floR*				*catA2*	
Chloramphenicol efflux pump							*cmlA5*	
Quaternary ammonium efflux pump		*qacEdelta1*	*qacEdelta1*	*qacEdelta1*			*qacEdelta1*	
Tetracycline		*tet(B)*		*tet(B)*			*tet(A)*	*tet(A)*
Fosmidomycin		*cyaA_S352T*	*cyaA_S352T*	*cyaA_S352T*				
Trimethoprim		*dfrA17*	*dfrA12*	*dfrA14*				
Macrolide		*mph(A)*	*mph(A) erm(B)*	*mph(A) erm(B)*				
Bleomycin				*ble*				
Colistin								*pmrB_R256G*

## Data Availability

Genomic sequences of *K. pneumoniae* isolates have been deposited in the BioProject database, (available online at http://www.ncbi.nlm.nih.gov/bioproject/632581, accessed on 1 July 2021); BioSample accessions including SAMN14913414, SAMN14913415, SAMN14913416, and SAMN14913417. Genomic sequences of *E. coli* isolates have been deposited in the BioProject database (available online at https://www.ncbi.nlm.nih.gov/sra/PRJNA630112, accessed on 1 July 2021); BioSample accessions including SAMN14943485, SAMN14943486, SAMN14943487, and SAMN14943488.

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
