# Peer review of "Whole Genome Sequence Analysis of Multidrug Resistant Escherichia coli and Klebsiella pneumoniae Strains in Kuwait"

_microorganisms, 2022, doi:10.3390/microorganisms10030507_

Round 1
Reviewer 1 Report
The authors herein presented a manuscript on the opportunity to use wgs for monitoring resistance issues in E.coli and k.pneumoniae in the context of MDR.
Although the manuscript is well arranged, material and methods properly describe the workflow and results are clearly presented, it appears similar to other works on the same topic. In my opinion the paper in this form do not provide any significant advance in this field.
I strongly suggest to authors to introduce other point of view including functional genomics as well expression data on it.
Author Response
February 16, 2022
Subject: manuscript microorganisms-1583233
Dear Editor,
Thank you for your letter and for giving us the opportunity to submit a revised draft of the manuscript. We appreciate the time and effort that the reviewers have dedicated to review and provide their comments on our manuscript. We have read all insightful comments by the reviewers carefully and addressed them point-by-point in the covering letter. Our responses are highlighted in bold below and all suggested modifications have been incorporated and highlighted in red within the revised manuscript.
Reviewer #1 comments to the authors:
The authors here in presented a manuscript on the opportunity to use wgs for monitoring resistance issues in E. coli and k. pneumoniae in the context of MDR.
Although the manuscript is well arranged, material and methods properly describe the workflow and results are clearly presented, it appears similar to other works on the same topic.
- “In my opinion the paper in this form do not provide any significant advance in this field. I strongly suggest to authors to introduce other point of view including functional genomics as well expression data on it”.
Author response: We would like to thank reviewer #1 for his/her constructive comments, which have helped us to substantially improve our manuscript. This study is the first study conducted on MDR carbapenem-resistant strains from healthy participants working in community settings in Kuwait at the whole genome level. Accordingly, this may enhance our preparedness to respond to emerging threats beyond healthcare settings. In response to the reviewer's comments, we have complied with the reviewer’s suggestion on functional genomics of the isolates and other changes were made. The modifications have been made in several paragraphs and highlighted in red in the revised manuscript.
Reviewer 2 Report
This manuscript describes 4 antibiotic resistant E. coli and 4 antibiotic resistant K. pneumoniae based on whole genome sequence data. In my opinion this study was well-designed and obtained results are novel and broaden our knowledge.
Comments
1) Title should be modified a little bit. I suggest this version: Whole genome sequence analysis of multidrug resistant 2 Escherichia coli and Klebsiella pneumoniae strains in Kuwait
2) What was the reason to include these 8 strains into this study? This should be described in the materials and methods part.
Author Response
February 16, 2022
Subject: manuscript microorganisms-1583233
Dear Ms. Bianca Ojovan,
Thank you for your letter and for giving us the opportunity to submit a revised draft of the manuscript. We appreciate the time and effort that the reviewers have dedicated to review and provide their comments on our manuscript. We have read all insightful comments by the reviewers carefully and addressed them point-by-point in the covering letter. Our responses are highlighted in bold below and all suggested modifications have been incorporated and highlighted in red within the revised manuscript.
Reviewer 2 comments to the authors:
This manuscript describes 4 antibiotic resistant E. coli and 4 antibiotic resistant K. pneumoniae based on whole genome sequence data. In my opinion this study was well-designed and obtained results are novel and broaden our knowledge.
- “Title should be modified a little bit. I suggest this version: Whole genome sequence analysis of multidrug resistant Escherichia coli and Klebsiella pneumoniae strains in Kuwait”.
Author response: We thank reviewer #2 for his/her kind words and for this positive feedback and suggestions. We have complied accordingly.
- “What was the reason to include these 8 strains into this study? This should be described in the materials and methods part”
Author response: Thank you for pointing this out to us. We performed whole genome sequencing on selected isolates for insight into the origin and mechanism of genetic diversity. We have inserted the detailed reason for including these isolates on this study on page 2 , lines 75-81.
We hope the revised manuscript meet your high standards and will better suit the journal “microorganisms”. The authors welcome further constructive comments if any. We thank you for your interest in our research.
Round 2
Reviewer 1 Report
The authors included interesting changes in this version. Data at the genomic level, well-distributed along with the text, provide sufficient background for enhancing the functionality of the manuscript.
Author Response
Author response: We thank reviewer #1 for his/her positive feedback and suggestions. We have complied accordingly. Suggested modifications have been incorporated and highlighted in blue within the revised manuscript, lines 62-64 page 2; lines: 295-296 page 8; lines 381-387; and 396-399 page 10.
The authors welcome further constructive comments if any.